# A Logical Framework for Verification of AI Fairness

## Abstract

With the widespread use of AI in socially important decision-making processes, it becomes crucial to ensure that AI-generated decisions do not reflect discrimination towards certain groups or populations. To address this challenge, our research introduces a theoretical framework based on the spider diagram—a reasoning system rooted in first-order predicate logic, and an extended version of the Euler and Venn diagrams—to define and verify the fairness of AI algorithms in decision-making. This framework compares the sets representing the actual outcome of the model and the expected outcome to identify bias in the model. The expected outcome of the model is calculated by considering the similarity score between the individual instances in the dataset. If the set of actual outcomes is a subset of the set of expected outcomes and all constant spiders in the former set have a corresponding foot in the expected outcome set, then the model is free from bias. We further evaluate the performance of the AI model using the spider diagram which replaces the conventional confusion matrix in the literature. The framework also permits us to define a degree of bias and evaluate the same for specific AI models. Experimental results indicate that this framework surpasses traditional approaches in efficiency, with improvements in processing time and a reduced number of function calls.

*Keywords: bias, discrimination, fairness, spider diagram, first-order predicate logic.*

## 1 Introduction

AI advancement raises fairness concerns and highlights the necessity of ethical and unbiased AI models (Mehrabi et al., 2021). It involves ensuring that an algorithm's predictions do not unfairly advantage or disadvantage particular subgroups of the population based on their demographic characteristics. Previous studies have illustrated numerous instances where AI systems have led to unintended biases. For example, Amazon developed an AI-based recruitment system that favored male candidates for technical job roles due to gender-biased data used in training the system (Dastin, 2018; Mujtaba & Mahapatra, 2019). Biases have also been observed in facial recognition systems, which often perform better for certain racial groups and poorly for others (Leslie, 2020). The US healthcare system was found to be racially biased, favoring white patients over African-American patients (Obermeyer et al., 2019; Feagin & Bennefield, 2014). Additionally, the COMPAS score, used in several states to assess inmate risk levels, was reported to be biased against African-American inmates based on certain fairness measures (Chouldechova, 2017; Berk et al., 2021). Consequently, there is a need for a standard procedure to verify and measure fairness in AI models (Buyl & De Bie, 2022; Richardson & Gilbert, 2021; Bellamy et al., 2018).

In the realm of evaluating fairness in an AI model, there are multiple approaches. These include statistical measures, individual fairness considerations, Fairness Through Unawareness (FTU), counterfactual or causal fairness, and logic-based approaches (Ignatiev et al., 2020). It's important to note that in the case of counterfactual fairness, a scenario where, for instance, the gender of an individual is hypothetically changed to a different value would lead to differences in other features as well. This complexity arises due to the interconnected nature between sensitive and non-sensitive attributes, making it challenging to accurately assess bias. Likewise, in the case of Fairness Through Unawareness (FTU), when certain features are linked or correlated with sensitive attributes, a model that overlooks these sensitive features doesn't guarantee fairness (Castelnovo et al., 2022). In our work,

we primarily focus on statistical fairness criteria. In the literature, statistical methods like predictive parity, demographic parity, equalized odds, disparate impact, equal opportunity, and statistical parity are used to verify fairness in machine learning models (Pessach & Shmueli, 2022; Lohia et al., 2019; Agarwal et al., 2023). Demographic parity, statistical parity, and Predictive parity refer to the idea that the proportion of positive predictions should be the same for all groups (Verma & Rubin, 2018) (Mehrabi et al., 2021) (Dwork et al., 2012) and do not take into account the fact that different groups may have different base rates or prior probabilities, which can lead to unequal treatment. Equalized odds mean that the true positive rate and false positive rate should be the same for all groups. This metric assumes that the costs of false positives and false negatives are equal, which may not be the case in all scenarios (Hardt et al., 2016). Disparate impact is a measure of whether a model is treating one group unfairly based on protected characteristics. However, this metric does not take into account the fact that certain features may be highly correlated with protected attributes, which can make it difficult to disentangle the impact of different factors on the final outcome (Friedler et al., 2014). Each of these metrics has its own strengths and weaknesses. They often lack transparency and have an overhead in terms of processing time and the number of function calls.

This study developed a theoretical framework based on spider diagrams (Howse et al., 2005; Stapleton et al., 2004) to define and verify fairness in AI. A detailed discussion on the spider diagrams is given in Section 2.1. This framework based on first-order predicate logic (FOPL) (Andréka et al., 2017; Kaldewaij, 1990; Barwise, 1977) first identifies $\alpha$ (discriminating) and $\omega$ (non-discriminating) classes from the input data set. It then calculates the expected outcome of every individual in the dataset. The expected outcome (ground truth) of the model is obtained by considering the similarity score between the individuals (how much alike the elements are, calculated based on the distance between the values of corresponding features). The actual outcome is the outcome given by the model's prediction. To verify fairness in the model, the framework compares the set of the expected outcomes ($\mathbb{G}$) to the set of the actual outcomes ($\mathbb{A}$). The model is fair if the former set is a subset of the latter and vice versa. Further, the bias can be visualized using the spider diagram which gives a higher-order object-level representation and replaces the conventional confusion metrics used to visualize the error(bias) and evaluate the performance of the model. The circles in the spider diagram represent sets $\mathbb{G}$ and $\mathbb{A}$. The intersection region of the circles gives the logical condition that corresponds to the equivalence of spiders in both set $\mathbb{G}$ and set $\mathbb{A}$. Using this condition, we define the fairness of an AI model (Definition 1). For this, the formal proof is explained in Theorem 1 in Section 3.1. By visualizing the bias using the spider diagram, the AI model's accuracy can be calculated by dividing the count of individuals in the intersection area of sets representing $\mathbb{G}$ and $\mathbb{A}$ with the total number of elements (individuals) in the given dataset. The formal proof is explained in Theorem 2 in Section 3.4. Additionally, the degree of bias for both $\alpha$ and $\omega$ classes of individuals in the dataset can be calculated using Algorithm 1 explained in Section 3.3. To find this, the frequency of occurrence of $\alpha$ and $\omega$ classes in both the actual outcome set ($\mathbb{A}$) and the expected outcome set ($\mathbb{G}$) is calculated and the detailed steps are shown in algorithm 1. It is found that compared to the existing methods, the new framework can be used in any data set and it precisely reasons the system's behavior. Experimental results show our new method to verify fairness is better by up to 95% in terms of processing time and 97.7% in terms of function calls (Table 1). Our main contributions are :

- We define FairAI, a framework based on the spider diagram to define and verify fairness (by visualizing bias) in an AI model (Section 3.1 and 3.2).
- We develop an algorithm 1 to calculate the degree of bias for discriminating (non-protected) and non-discriminating (protected) classes (groups) of individuals in an AI model.
- We demonstrate that the use of spider diagrams for visualizing bias (errors) is more optimized than the confusion matrix in terms of processing time and function calls (Table 2).
- Further, we show that the spider diagram can be used to measure the performance of the model (accuracy, precision, recall, sensitivity, specificity, etc.,) (Theorem 3).

The paper is structured as follows: Section 2, includes the formal definition of AI model used in this paper, and a brief overview of calculating the expected outcome of an AI model. In Section 2.1, we provide an overview of the spider diagram. In Sections 3.1 and 3.2, we describe the methodology used to define and verify fairness in an AI model based on the spider diagram (formal theorem with proof). In Section 3.3, we discuss the Algorithm 1 to find the degree of bias of an AI model. In Section 3.4, we briefly describe FOPL formulas for the performance evaluation of an AI model. Experimental results are discussed in Section 4. In Section 5, we conclude with our findings.

## 2 PRELIMINARIES

In this study, we use an AI model defined by (Das & Rad, 2020) with two demographic groups (subpopulation)—non-protected ($\alpha$) and protected ($\omega$)—based on the sensitive attribute(s) (race, age, gender) identified from the dataset. This work considers sensitive attributes to ensure that the algorithm does not discriminate against certain groups. The scope of this work is on the classification models. For example, if an algorithm is used to identify loan defaulters, using a sensitive attribute like gender or age can help to ensure that the algorithm does not unfairly discriminate against women or the elderly. We use $\omega$ and $\alpha$ to represent protected and non-protected groups of individuals in an AI model (based on the sensitive attribute(s) like race, age, sex, etc.,). Protected groups are advantaged and non-protected groups are disadvantaged by the algorithm's decision. This is an assumption that aligns with past experiences (Chouldechova, 2017; Dastin, 2018). We use a dataset $D$ containing $N$ data points (instances), where each instance is a tuple $\langle \mathbf{x} \in \mathbb{R}^K, y \in \{0, 1\}, \hat{y} \in \{0, 1\}\rangle$ representing the input vector, expected outcome label, and the actual outcome label respectively. i.e., $D = \{(x_n, y_n, \hat{y}_n)\}_{n=1}^N$. We use the notation $x_i$ to denote an individual in $D$ and $y_i$ to denote the corresponding outcome. Further, we have a function (model) $M$ that predicts the outcome y given input x. i.e., $M : x \rightarrow y$. In this paper, we introduce the notations $\mathbb{G}$ and $\mathbb{A}$ to denote the sets of expected and actual outcomes respectively, produced by an AI model. We use the term sensitive or protected attributes interchangeably in this work. We use the terms instance, entity, and individual interchangeably. In this paper, we use the symbol $\#$ to represent the count of individuals in the sets $\mathbb{G}$ and $\mathbb{A}$.

In this work, the expected outcome of the model is obtained by considering the similarity score between the individuals in the dataset. To find this, our method picks a random data point from the dataset as a generator. For finding the similarity, the distance between a point (instance in $D$) and the generator can be calculated. In this paper, we use Euclidean distance, also called the $L^2$-norm to calculate similarity metrics using (1). Compared to other distance metrics in literature, Euclidean distance provides a straightforward geometric interpretation and can be used in various data scales due to scale invariance. It preserves geometric features for continuous data, which makes it a commonly used metric in numerous fields due to its simplicity and versatility (Alpaydin, 2014). Consider two points $Q_i$ and $Q_j$ and a generator $Q_1$. $Q_i$ maps to the set of $Q_1$ (i.e., the outcome of $Q_i$ and $Q_1$ are equal), if the distance between itself and $Q_1$ is less than the distance from $Q_j$ to $Q_1$ (Okabe & Suzuki, 1997). Let $a_1, a_2, \ldots, a_m$ be the attributes, that include both sensitive attributes (i.e. race, ethnicity, sex) and non-sensitive attributes in the model. Here $m$ denotes the total number of attributes in the model. Calculate the distance between two different entities/individuals using the formula given below:

$$\Delta(Q_1, Q_i) = \sqrt{\sum_{i=1}^{N} \sum_{j=1}^{m} (Q_1^{a_j} - Q_i^{a_j})^2} \tag{1}$$

In (1), $Q_1$ is taken as a generator point and $Q_i$ corresponds to an entity in the data set. The shorter the distance between entities, the similarity between them increases. We can say if two entities $Q_1$ and $Q_i$ are similar, the expected outcome of both given the set of attributes $a_1, a_2, ...a_m$ should be the same. i.e. they both should be classified into the same class label (0 or 1) depending on the class of $Q_1$. If we repeat this for all rows of entities, then the similarity between them is calculated. Hence, we get the expected outcome for all instances in $D$. The entities/individuals (instances) with equal merits (similarity score) get equal output and entities with different merit (similarity score) proportions will be mapped to different output classes. For doing this we assume a threshold value $t$ which is assumed to be equal to the average of Euclidean distance calculated for each data point. If the value of similarity between $Q_1$ and $Q_i$ from (1) is less than or equal to the threshold, then $Q_i$ should be mapped to the output class of $Q_1$ otherwise to the other class. In the context of AI models, bias refers to the presence of systematic errors or inaccuracies in the way the model operates. Bias can arise from a number of factors, including the data used to train the model, the algorithms and assumptions used in the model, and the way the model is designed and implemented (Bellamy et al., 2018). This can result in the model making decisions or predictions that are unfair or discriminatory towards certain groups of individuals, based on factors such as race, gender, or socioeconomic status. Addressing bias is an important area of research in the field of AI and machine learning (Mehrabi et al., 2021). In this paper, we use a logical reasoning system based on first-order predicate logic (FOPL) called spider diagrams to define fairness and visualize bias. FOPL provides a formal and precise representation of knowledge and relationships (Manna, 1969; Van Emden & Kowalski, 1976).

It allows for clear and unambiguous statements about the relationships between entities and their properties. This is crucial in applications where understanding the reasoning process and transparency is important.

## 2.1 SPIDER DIAGRAMS

Spider diagrams are higher-order representations of Venn diagrams and are equivalent to monadic FOL with equality. Here each closed curve is used to represent a set and is labeled and enclosed in a rectangle. The sets are divided into regions. A basic region is a region in the plane enclosed by a curve. The intersection, union, and difference of the two regions will be a region provided it is non-empty. A zone is a region having no other regions in it. Each zone contains a group of spiders (denotes the existence of elements in the set) residing in it. A spider is a plane tree with vertices (feet) placed in different zones and edges (line segments or tie) are used to connect between them (Howse et al., 2005). If one of the feet (vertex) of the spider

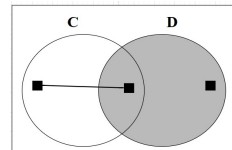

Figure 1: Spider diagram example.

is in a zone, it implies the spider touches the zone and resides in it and is called the habitat of the spider. Zones can be shaded or unshaded. The shading of zones represents an empty set if there is no spider residing in it, otherwise, this gives the upper bound of cardinality. Each spider denotes distinct individuals and if they are connected by a tie (edge), they both represent the same individual (Fish & Flower, 2005; Stapleton & Howse, 2006; Gil et al., 1999; Delaney et al., 2008; Stapleton et al., 2009). This implies that the region they both reside (habitat) will be the same. If there are two constant spider $X$ and $Y$ habitats of the region $r_1$ and $r_2$ respectively are connected with a tie, then both $X$ and $Y$ represent the same individual (element), and hence $r_1 = r_2$. Spiders can be of different types like existential spiders (used to represent existential quantization, i.e., there exists), universal spiders (used to represent universal quantification, i.e., for all), and constant spiders (used to represent individuals in a relation). We recommend that interested readers to refer (Howse et al., 2005; Stapleton & Howse, 2006) for further details. Figure 1 presents an example of a spider diagram. It contains two sets labeled as $\mathbb{C}$ and $\mathbb{D}$ with four zones. Two zones are shaded. The diagram contains two spiders—a single-footed spider inside the zone $\mathbb{D} \setminus \mathbb{C}$ and a two-footed spider residing in zone $\mathbb{C} \setminus \mathbb{D}$ or $\mathbb{C} \cap \mathbb{D}$ (i.e., completely inside the set $\mathbb{C}$).

## 3 METHODOLOGY

### 3.1 FRAMEWORK FOR FAIR AI BASED ON SPIDER DIAGRAMS

AI fairness refers to the extent to which an artificial intelligence system treats all individuals or groups fairly and equitably, without creating or perpetuating biases or discrimination. Achieving this requires identifying and addressing sources of bias in the data used to train the algorithm, designing and testing the algorithm to detect and mitigate potential biases, and evaluating the algorithm's performance on various fairness metrics (Lohia et al., 2019). Here, we formally define AI fairness using a logical condition based on the spider diagrams.

**Definition 1.** *The AI model $M$ represented using a constant spider diagram is fair for a dataset $D$, set of actual outcomes $\mathbb{A}$ and set of expected outcomes $\mathbb{G}$, if $\forall g_i \in \mathbb{G}(\exists a_i \in \mathbb{A} \implies g_i = a_i)$, where $g_i$ and $a_i$ are the expected and actual outcome of an individual $x_i$.*

An AI model is unbiased if it satisfies the properties of fairness, transparency, and explainability. In this research, we focus on fairness to ensure the model is unbiased. Theorem 1 gives formal proof for the same. If the model is fair, then all spiders lie inside the intersection area of the sets $\mathbb{G}$ (set of the expected outcome of the model) and $\mathbb{A}$ (set of the actual outcome of the model) and the set representing the actual outcomes should be a subset of the expected outcomes. The zone $\{\{\mathbb{G} \cap \mathbb{A}\}, \phi\}$ represents the logical condition that the spider is residing only in the intersection region of $\mathbb{G}$ and $\mathbb{A}$ and not outside it. Hence in an AI model, if all spiders lie inside the zone $\{\{\mathbb{G} \cap \mathbb{A}\}, \phi\}$, then the model is unbiased. Demonstration of the working of the spider diagram for bias visualization and fairness verification is explained in Figure 7 given in Appendix B.

**Theorem 1** (Fair AI). *An AI model is unbiased if all spiders lie inside the zone $\{\{\mathbb{G} \cap \mathbb{A}\}, \phi\}$. (Proof in Appendix A)*

### 3.2 BIAS VISUALIZATION USING SPIDER DIAGRAMS

In this work, we use spider diagrams with constants to visualize bias in an AI model. We use the formal definition of a constant spider diagram from (Stapleton et al., 2013) to represent model $M$ as shown below.

**Definition 2.** *A constant spider diagram used to represent an AI model $M$ is a tuple $\langle L, S, Z, f, R \rangle$ where:*
$L \in \{\mathbb{A}, \mathbb{G}\}$, *is a finite set of labels where* $\mathbb{G} = \{y_i \in \{0, 1\}$, *finite set of expected outcome} and* $\mathbb{A} = \{\hat{y}_i \in \{0, 1\}$ *is a finite set of actual outcome}*
$S$ *is a set of constant spider labels representing individuals, i.e.,* $\{(g_i, a_i) : g_i \in \mathbb{G}$ *and* $a_i \in \mathbb{A}, i = 1, \ldots, N\}$
$Z \subseteq \{(z, L \setminus z) : z \subseteq L\}$ *is a finite set of zones inside $z$ and outside $L \setminus z$.*
$f : S \to Z$ *is a function that maps each constant spider to a zone. If $f(a_i) = z \in Z$, then $a_i$ resides in zone $z$.*
$R \in Z \setminus \{\emptyset\}$ *is the set of regions, where $\emptyset$ denotes the empty set.*

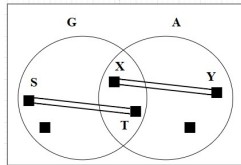
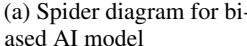
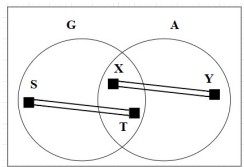

(a) Spider diagram for biased AI model      (b) Spider diagram for unbiased AI model

Figure 2: Spider diagrams

Spider diagrams help to compare the actual and expected outcome of the $\alpha$ and $\omega$ classes visually as shown in the above figure. Fig 2a and 2b represent spider diagrams for a biased and unbiased model respectively. It shows the mapping of elements in the actual outcome ($\mathbb{A}$) of $\omega$ and $\alpha$ classes to the expected outcome ($\mathbb{G}$) set. If two spiders $X$ and $Y$ are connected by a tie then, it gives the mapping of spiders into a zone. Here $Y \to X$ mapping represents constant spider $Y$ in set $\mathbb{A}$ is connected with a tie to constant spider $X$ in set $\mathbb{G}$ or $\mathbb{G} \cap \mathbb{A}$. Hence, $X$ and $Y$ are the two feet of a spider i.e., $X = Y$. Similarly, $S \to T$ mapping represents spider $S$ is connected to spider $T$ from set $\mathbb{G}$ to $\mathbb{A}$ or $\mathbb{G} \cap \mathbb{A}$. If an AI model $M$ can be represented using this diagram, and if it follows the logical constraint $(\forall S \in \mathbb{G}(\exists T \in \mathbb{A}(S = f.T))) \land (\forall Y \in \mathbb{A}(\exists X \in \mathbb{G}(Y = f.X)))$, where $f$ is a mapping function which maps the spider to the zone it belongs to, then model $M$ is free from bias. Fig 2a represents a spider diagram for a biased AI model where the logical condition does not hold true and there exist some spiders in sets $\mathbb{G}$ and $\mathbb{A}$ which are not connected by a tie. This indicates that there are some elements that are not common across these sets. On the other hand, Fig 2b holds the logical condition for an unbiased AI model as no spiders lie outside the intersection area.

Using the spider diagram for bias visualization helps us to replace the confusion matrix used to measure and evaluate the performance of an AI model. Confusion matrix may sometimes lead to misclassification especially when the number of output classes increases. In such cases, False Negatives (FN) for one class can be False Positives (FP) for another (Beauxis-Aussalet & Hardman, 2014). Also, to fully understand the error rates, the user must go through the matrix's columns and rows. Yet, this examination provides only a high-level overview of the errors and doesn't shed light on the distribution of $\alpha$ and $\omega$ groups within each error categories. Hence in our work, we replace the confusion matrix with the spider diagram. This method of bias visualization helps to reduce the ambiguity in using confusion matrices and gives a detailed bias visualization with reduced function calls and processing time as shown in Figure 3.

### 3.3 DEGREE OF BIAS FOR $\alpha$ AND $\omega$ CLASSES IN AN AI MODEL

Algorithm 1 describes the method to find the degree of bias for $\alpha$ and $\omega$ classes in the model based on their occurrence frequency in the actual and expected outcome. The degree of bias for the groups (classes) of individuals is a number between +1 and –1 with 0 meaning no bias. 0 to +1 indicates a

bias towards the group (i.e., the model is favoring the group) and 0 to –1 indicates a bias against the group (i.e., the model is not favoring the group). In this paper, the $\omega$ class of individuals (which is the protected group) will always have a degree of bias 0 or a value between 0 and +1 whereas, the $\alpha$ class of individuals (non-protected group) always have a degree of bias between –1 to 0. The detailed experimental values of the degree of bias for various datasets used in this paper are shown in Fig 5 in section 4.

---

**Algorithm 1** Degree of bias for $\alpha$ and $\omega$ classes in an AI model

---

1. **Input:** $\mathbb{A}$: set of actual outcome, $\mathbb{G}$: set of expected outcome
2. **Output:** Degree of bias ($d$) of classes $\alpha$ and $\omega$ in sets $\mathbb{A}$ and $\mathbb{G}$.
3. $int\ i,\ N \geq 0$   /*i is the loop variable,$N$ represents the number of individuals in dataset*/
4. $initialize\ i, A_\alpha, A_\omega, G_\alpha, G_\omega, d_\alpha, d_\omega \leftarrow 0$
5. $A_\alpha, A_\omega \leftarrow getfrequency(\mathbb{A})$ /* stores the frequency of $\alpha$ and $\omega$ classes in actual outcome set. */
6. $G_\alpha, G_\omega \leftarrow getfrequency(\mathbb{G})$  /* stores the frequency of $\alpha$ and $\omega$ classes in expected outcome set. */
7. $d_\alpha \leftarrow \frac{(A_\alpha - G_\alpha)}{(\#\alpha)}$ /*$d_\alpha$ stores the value of degree of bias for $\alpha$ class in the model*/
8. $d_\omega \leftarrow \frac{(A_\omega - G_\omega)}{(\#\omega)}$ /*$d_\omega$ stores the value of degree of bias for $\omega$ class in the model*/
9. $getfrequency(\mathbb{B})$ /* function definition using $\mathbb{B}$ as formal parameter. */
10. **while** ( i $\neq N$) **do**
11.     **if**($\mathbb{B} \neq \emptyset$) **then**
12.         **if**($\alpha \in \mathbb{B}$) **then**
13.            $B_\alpha \leftarrow B_\alpha + 1$  /* increments $B_\alpha$ if output is $\alpha$ */
14.         **else** $B_\omega \leftarrow B_\omega + 1$   /* increments $B_\omega$ if output is $\omega$ */
15.         **end if**
16.     **end if**
17.   $i \leftarrow i + 1$      /*increments $i$ by 1 */
18. **end while**
19. **return** $B_\alpha, B_\omega$
20. **return** $d_\alpha, d_\omega$

---

In Algorithm 1, line 3 signifies the declaration of two variables: $i$ which serves as the loop variable, and $N$, which stores the total number of records in the dataset. Line 4 initializes the following variables to zero: $i$, $A_\alpha$, $A_\omega$, $G_\alpha$, $G_\omega$, $d_\alpha$, and $d_\omega$. In this context, $A_\alpha$ and $A_\omega$ are used to store the frequency of occurrences of $\alpha$ and $\omega$ within the actual outcome set, while $G_\alpha$ and $G_\omega$ are used for the same function for the expected outcome set. The variables $d_\alpha$ and $d_\omega$ are used to store the extent of bias associated with the $\alpha$ and $\omega$ classes in the model denoted as $M$. Line 5 calculates the frequency of $\alpha$ and $\omega$ within the set $\mathbb{A}$, while Line 6 performs a similar calculation for the set $\mathbb{G}$. Lines 7 and 8 are responsible for determining the bias degree ($d$) for both the $\alpha$ and $\omega$ classes within an AI model using the formula specified in the algorithm. Here, $\#(\alpha)$ and $\#(\omega)$ represent the total count of occurrences of the $\alpha$ and $\omega$ groups in the input dataset. Lines 9–19 are the logic of the function to calculate the count of positive predictions (frequency of occurrence) of $\alpha$ and $\omega$ in set $\mathbb{A}$ and $\mathbb{G}$. In line 10, if the condition holds true after substituting the value of $i$ for the current iteration, lines 11–17 will be executed. Line 11 verifies whether the set $\mathbb{B}$ is not empty, while line 12 checks whether the value in $\mathbb{B}$ for the current iteration corresponds to $\alpha$. If this condition holds true, line 13 is executed, leading to an increment in the value of the variable $B_\alpha$. Conversely, if the value is $\omega$, line 14 is executed, incrementing the variable $B_\omega$ by 1. Subsequently, in line 17, the value of $i$ is incremented, and the control returns to line 10, repeating this process until $i$ reaches $N$. Once

the condition is no longer satisfied, the algorithm returns the values of $B_\alpha$ and $B_\omega$. Finally, line 20 returns the degree of bias values, $d_\alpha$ and $d_\omega$.

### 3.4 AI MODEL'S PERFORMANCE METRICS IN SPIDER DIAGRAMS: FORMULA INTERPRETATION

(Sokolova et al., 2006) calculates accuracy as the number of correct predictions, i.e., true positives and true negatives divided by the total number of predictions. However, in this work, spider diagrams are used to evaluate the performance of the model. Hence a new formula has been proposed to calculate accuracy. According to this, the accuracy of an AI model can be calculated by dividing the count of individuals in the intersection area of sets representing $\mathbb{G}$ and $\mathbb{A}$ in the spider diagram with the total number of elements (individuals) in the given dataset. The formal proof is as shown in Theorem 2. Additionally, the logical framework explained in this paper provides a way to describe the performance metrics of an AI model using formulas based on the spider diagrams. The formulas are obtained based on Theorem 3.

**Theorem 2** (AI accuracy). *The accuracy of an AI model M represented using a constant spider diagram can be expressed as the frequency of spiders residing in zone $\{\{\mathbb{G} \cap \mathbb{A}\}, \phi\}$.*

**Theorem 3.** *For an AI model M represented using a constant spider diagram, the sensitivity, specificity, and precision can be represented using the formulas as follows:*

$$Sensitivity = \frac{\#g \in \mathbb{G} \cap \mathbb{A} \mid g = 1}{\#g \in \mathbb{G} \mid g = 1}$$

$$Specificity = \frac{\#g \in \mathbb{G} \cap \mathbb{A} \mid g = 0}{\#g \in \mathbb{G} \mid g = 0}$$

$$Precision = \frac{\#g \in \mathbb{G} \cap \mathbb{A} \mid g = 1}{\#a \in \mathbb{A} \mid a = 1}$$

*Where g and a represents each individual in the sets $\mathbb{G}$ and $\mathbb{A}$.*

Proofs of Theorems 2 and 3 are given in Appendix A.

## 4 EXPERIMENTS

This paper introduces a theoretical framework for verifying fairness in an AI model. The key goal of this framework is to ensure that similar people are treated similarly by a classification model. The framework is tested using five datasets – social network ads prediction, loan approval prediction (dat, c), German credit score prediction (dat, b), UCI adult (dat, a) and US faculty hiring (dat, d) (Wapman et al., 2022) named as D1, D2, D3, D4, and D5 respectively in this paper (Please refer Appendix D for further experimental details). The experimental results show that the proposed method to verify fairness ( given in Theorem 1 in Section 3.1) is more optimized in terms of processing time and the number of function calls compared to the existing approaches including Demographic Parity (DP) (Dwork et al., 2012), Predictive Parity (PP) (Verma & Rubin, 2018) and Equalized Odds (EO) (Hardt et al., 2016). Table 1 shows that the new method is optimized up to 95 % in terms of processing time and 97.7 % in terms of the number of function calls.

Table 1: Performance comparison of EO, PP, DP with Fair AI to verify fairness

| Dataset | EO | | PP | | DP | | Fair AI | | % Improvement | |
| --- | --- | --- | --- | --- | --- | --- | --- | --- | --- | --- |
| | Function calls | Time (ms) | Function calls | Time (ms) | Function calls | Time (ms) | Function calls | Time (ms) | Time | Function calls |
| D1 | 3487 | 12 | 3830 | 12 | 1783 | 5 | 263 | 1 | 96.5 | 97.1 |
| D2 | 3226 | 9 | 3811 | 15 | 1522 | 8 | 263 | 2 | 93.7 | 96.9 |
| D3 | 3589 | 8 | 3966 | 12 | 1953 | 4 | 111 | 3 | 87.5 | 98.8 |
| D4 | 3226 | 25 | 3827 | 44 | 1522 | 24 | 177 | 1 | 98.8 | 97.9 |
| D5 | 4866 | 339 | 5630 | 142 | 2372 | 142 | 263 | 2 | 99.6 | 97.9 |

Figure 3 presents the visual depiction of bias within the models for various datasets calculated using the spider diagrams. Here we use stacked bar graphs to represent the results for better visualization. These graphs offer insights into the distribution of classification metrics (specifically, False Positives (FP), False Negatives (FN), and Correct Predictions labeled as CP which includes both True positives (TP) and True Negatives (TN)) across different classes ($\alpha$ and $\omega$) in various datasets. For instance, let's consider dataset D5 (US faculty hiring that discriminates against women candidates): In this dataset, the stacked bar graph makes the model's bias evident. It indicates that around 75.7% of instances from the class labeled as $\alpha$ receive False Negative predictions, while only 24.2% of instances from the class labeled as $\omega$ experience False Negatives. Additionally, for False Positive predictions, about 78.4% of instances from class $\omega$ are favored, whereas class $\alpha$ only sees 21.5%. This analysis unmistakably reveals the model's inclination towards favoring class $\omega$ over class $\alpha$. This interpretation can be applied to the other datasets to visualize bias in a similar manner.

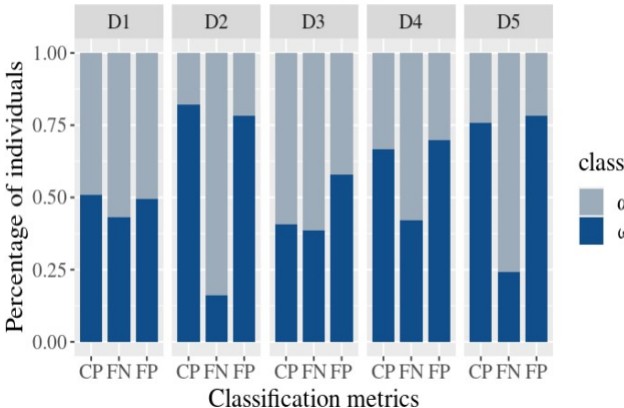

Figure 3: Bias visualization of various datasets using spider diagram.

Table 2 presents the experimental result for comparison of the spider diagram and confusion matrix. This shows that the performance evaluation using the spider diagram is more optimized in terms of processing time and the number of function calls than the confusion matrix. Here the number of recursion or function calls can be crucial in assessing a model's performance for a few reasons. Firstly, it indicates the computational load and efficiency of the model. A high number of recursive calls can suggest increased computational complexity, potentially leading to longer processing times or resource-intensive operations (Grzeszczyk, 2018; Ousterhout; Asadi et al., 2013). Graphical visualization of the results are shown in Figures 4a and 4b.

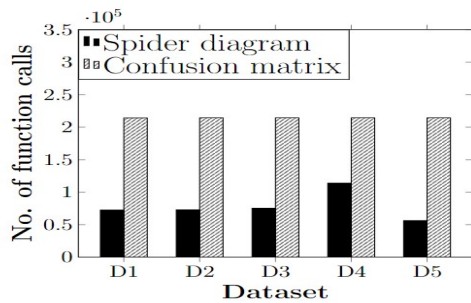

(a) Spider diagram vs confusion matrix in terms of number of function calls

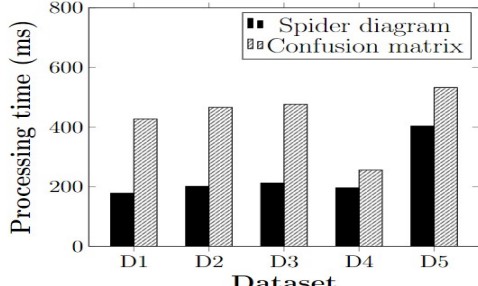

(b) Spider diagram vs confusion matrix in terms of processing time

Figure 4: Spider diagram vs confusion matrix

Table 2: Performance comparison of confusion matrix and spider diagram

| | Confusion matrix | | Spider diagram | | % Improvement | |
| Dataset | Function calls | Time (ms) | Function calls | Time (ms) | Time | Function calls |
| --- | --- | --- | --- | --- | --- | --- |
| D1 | 214031 | 427 | 72488 | 178 | 58.3 | 66.1 |
| D2 | 214146 | 466 | 72641 | 201 | 56.8 | 66.0 |
| D3 | 214155 | 476 | 75242 | 212 | 55.4 | 64.8 |
| D4 | 214155 | 256 | 113772 | 196 | 23.4 | 46.8 |
| D5 | 214227 | 532 | 56200 | 403 | 24.3 | 73.7 |

In Section 3, we presented an algorithm to calculate the degrees of bias in an AI model for classes $\alpha$ and $\omega$. The degrees of bias for various datasets are calculated and the results are given in Figure 5. As discussed in Subsection 3.3, the degree of bias value for a class ranges from -1 to +1, with 0 meaning no bias. A value from -1 to 0 indicates the algorithm is biased *against* the particular class in the model, and a value from 0 to +1 indicates a bias *towards* the class. As an example, for dataset D5, the model is biased *against* $\alpha$ class with a degree of $-0.96$ and biased *towards* $\omega$ class with a value of $+0.4$. Here biased *against* means disproportionately disadvantaged and biased *towards* means advantaged by the model.

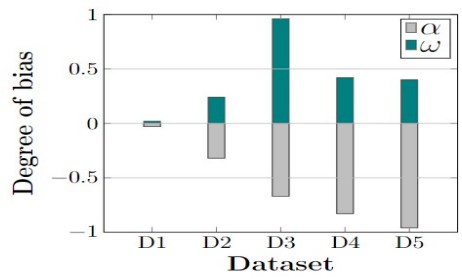

Figure 5: Degree of bias for $\alpha$ and $\omega$ class in the datasets

In Section 3.4, we present Theorem 2 to calculate the accuracy of an AI model using a formula based on the spider diagram. Figure 6 represents the optimization of the proposed formula 2 compared to the conventional method (Sokolova et al., 2006) used to calculate accuracy. The results show that the processing time of the proposed formula is nearly constant irrespective of the size of the dataset whereas, in the conventional method as the size of the dataset increases the processing time also increases linearly. Figure 8 in Appendix F illustrates the relationship between accuracy and fairness of the proposed method across 5 distinct datasets.

## 5 CONCLUSION

This paper presents a logical framework for defining and verifying the fairness of an AI model, along with an approach for evaluating model performance that serves as an alternative to the traditional confusion matrix. Additionally, this method introduces a more formal and precise way of expressing accuracy, specificity, and sensitivity formulas using the spider diagram, which adds a new dimension to the representation of these concepts. This framework also permits us to define the degree of bias and calculate the same to detect bias in the AI model. As shown in our experimental results, compared to existing approaches in the literature, the new framework is designed to detect potential bias with fewer function calls and in less processing time and hence improves the efficiency. The experimental findings showed that replacing the confusion matrix with the spider diagram for performance evaluation of the model offers better performance in terms of reduced processing time and function calls.

Figure 6: Performance comparison of AI accuracy 2 formula with the conventional formula (Sokolova et al., 2006) based on confusion matrix

Given the importance of having a general framework for verifying the discrimination of AI models against particular groups in society, this work can be used to formulate a logical condition to ensure fairness in such models. Currently, the framework is tested on a few datasets which are limited in the number of records. While this is sufficient to reach a conclusion about the framework's performance, certain flaws will become evident when tested on a large-scale dataset. In future work, we can plan to measure the performance and stability of the model when tested on a large dataset. Adding to this, the study can be extended to investigate accuracy-fairness tradeoff in case of dataset bias. Also, this framework can be extended further to incorporate other crucial considerations such as privacy and explainability.

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

APPENDIX

## A PROOFS OF THEOREMS

**Theorem A.1** (Fair AI). *An AI model is unbiased if all spiders lie inside the zone $\{\{\mathbb{G} \cap \mathbb{A}\}, \phi\}$.*

*Proof.* Suppose all spiders lie inside the zone $\{\{\mathbb{G} \cap \mathbb{A}\}, \phi\}$, then take a random spider $g_i \in \mathbb{G}$ .Since all spiders lie in $\{\{\mathbb{G} \cap \mathbb{A}\}, \phi\}$,

$$g_i \in \mathbb{G} \cap \mathbb{A} \implies g_i \in \mathbb{G} \wedge g_i \in \mathbb{A}.$$

Similarly $a_i \in \mathbb{A}$ lie inside $\{\{\mathbb{G} \cap \mathbb{A}\}, \phi\}$.

$$\implies a_i \in \mathbb{A} \wedge a_i \in \mathbb{G} \equiv \mathbb{G} \subseteq \mathbb{A} \wedge \mathbb{A} \subseteq \mathbb{G}$$

When we write this in first-order logic we get,

$$\forall g_i \in \mathbb{G}(\exists a_i \in \mathbb{A} \implies g_i = a_i) \wedge \forall a_i \in \mathbb{A}(\exists g_i \in \mathbb{G} \implies g_i = a_i)$$
$$\equiv \forall g_i \in \mathbb{G}(\exists a_i \in \mathbb{A} \implies g_i = a_i).$$

This shows that in an AI model $M$ represented using spider diagram if all spiders lie inside the zone $\{\{\mathbb{G} \cap \mathbb{A}\}, \phi\}$, then the model is fair and hence concluded as unbiased. $\square$

**Theorem A.2** (AI Accuracy). *The accuracy of an AI model M represented using a constant spider diagram can be expressed as the frequency of spiders residing in zone $\{\{\mathbb{G} \cap \mathbb{A}\}, \phi\}$.*

*Proof.* The statistical formula for accuracy is given by $\frac{(TP+TN)}{N}$ where $N$ is the total number of predictions in $M$, $TP$ represents true positive outcomes and $TN$ represents true negative outcomes. They can be represented in first-order logic as,

$$TP = (\#i : 0 \le i \le N : g_i = 1 \wedge a_i = 1)$$
$$TN = (\#i : 0 \le i \le N : g_i = 0 \wedge a_i = 0)$$

where $g_i$ and $a_i$ are spiders representing individual $i$ residing in sets $\mathbb{G}$ and $\mathbb{A}$ respectively.

$$TP + TN = (\#i : 0 \le i \le N : g_i = a_i)$$

This gives the count of spiders where $g_i = a_i$. For $g_i$ to be equal to $a_i$ in the spider diagram, $g_i$ should be connected by a tie to $a_i$ and both should lie in the same zone. $\implies TP + TN = \#(\mathbb{G} \cap \mathbb{A})$. Hence the accuracy can be given by $\frac{\#(\mathbb{G} \cap \mathbb{A})}{N}$ which gives the frequency of spiders residing in the zone $\{\{\mathbb{G} \cap \mathbb{A}\}, \phi\}$ $\square$

**Theorem A.3.** *For an AI model M represented using a constant spider diagram, the sensitivity, specificity, and precision can be represented using the formulas as follows:*

$$Sensitivity = \frac{\#g \in \mathbb{G} \cap \mathbb{A} \mid g = 1}{\#g \in \mathbb{G} \mid g = 1}$$
$$Specificity = \frac{\#g \in \mathbb{G} \cap \mathbb{A} \mid g = 0}{\#g \in \mathbb{G} \mid g = 0}$$
$$Precision = \frac{\#g \in \mathbb{G} \cap \mathbb{A} \mid g = 1}{\#a \in \mathbb{A} \mid a = 1}$$

*Where $g$ and $a$ represents each individual in the sets $\mathbb{G}$ and $\mathbb{A}$.*

*Proof.* True Positives ($TP$), True Negatives ($TN$), False Positives ($FP$), and False Negatives ($FN$) can be represented in First-order logic as shown below. True positives, in the context of binary classification models, refer to the instances or data points that are correctly identified as belonging to the positive class by the model. In other words, a true positive occurs when the model's actual outcome is positive (i.e., 1) and matches the expected outcome, which is also positive (i.e., 1). When the expected outcome is positive for an individual $i$ within the dataset $D$, it can be represented as $g_i = 1$. Similarly, when the actual outcome is positive, it can be denoted as $a_i = 1$. Using these representations, we can formulate the FOPL (First-Order Predicate Logic) formula for True Positives as demonstrated below. This approach can be extended to derive formulas for True Negatives, False Negatives, and False Positives, respectively.

$$TP = (\#i : 0 \le i \le N : g_i = 1 \wedge a_i = 1)$$
$$TN = (\#i : 0 \le i \le N : g_i = 0 \wedge a_i = 0)$$
$$FP = (\#i : 0 \le i \le N : g_i = 0 \wedge a_i = 1)$$
$$FN = (\#i : 0 \le i \le N : g_i = 1 \wedge a_i = 0)$$

Here, $g_i$ and $a_i$ are the spiders representing individual $i$ in the sets $\mathbb{G}$ and $\mathbb{A}$ respectively. $N$ is the total number of instances in the dataset. The symbol $\#i$ denotes the number of individuals for which the specified condition is satisfied.

**Case 1.** *(Sensitivity)— The statistical formula for sensitivity is $\frac{TP}{TP+FN}$.*
$TP = (\#i : 0 \le i \le N : g_i = 1 \wedge a_i = 1) \equiv (\#g \in \mathbb{G} \cap \mathbb{A} \mid g = 1)$
$TP + FN = (\#i : 0 \le i \le N : (g_i = 1 \wedge a_i = 1) \vee (g_i = 1 \wedge a_i = 0))$
$\equiv (\#g \in \mathbb{G} \mid g = 1)$
*Hence, Sensitivity =* $\frac{\#g \in \mathbb{G} \cap \mathbb{A} \mid g = 1}{\#g \in \mathbb{G} \mid g = 1}$

**Case 2.** *(Specificity)—The statistical formula for specificity is $\frac{TN}{TN+FP}$.*
$TN = (\#i : 0 \le i \le N : g_i = 0 \wedge a_i = 0) \equiv (\#g \in \mathbb{G} \cap \mathbb{A} \mid g = 0)$
$TN + FP = (\#i : 0 \le i \le N : (g_i = 0 \wedge a_i = 0) \vee (g_i = 0 \wedge a_i = 1))$
$\equiv (\#g \in \mathbb{G} \mid g = 0)$
*Hence, Specificity =* $\frac{\#g \in \mathbb{G} \cap \mathbb{A} \mid g = 0}{\#g \in \mathbb{G} \mid g = 0}$

**Case 3.** *(Precision)—The statistical formula for precision is $\frac{TP}{TP+FP}$.*
$TP = (\#i : 0 \le i \le N : g_i = 1 \wedge a_i = 1) \equiv (\#g \in \mathbb{G} \cap \mathbb{A} \mid g = 1)$
$TP + FP = (\#i : 0 \le i \le N : (g_i = 1 \wedge a_i = 1) \vee (g_i = 0 \wedge a_i = 1))$
$\equiv (\#a \in \mathbb{A} \mid a = 1)$
*Hence, Precision =* $\frac{\#g \in \mathbb{G} \cap \mathbb{A} \mid g = 1}{\#a \in \mathbb{A} \mid a = 1}$

$\square$

## B    DEMONSTRATION OF THE METHODOLOGY USING A TOY EXAMPLE

Figure 7 demonstrates the application of spider diagrams in assessing AI models. This example is constructed from a small dataset and a biased model. In the context of classification, we introduce a simplified example involving two subgroups within a population distinguished by protected attributes (blue and red), representing women and men respectively. In the top left set $\mathbb{G}$, 'x' denotes positive ground truth. The outcomes of the AI model are presented in the bottom set $\mathbb{A}$ on the left side, with '+' marks indicating positive predictions. If an individual falls within the protected attribute category "blue" and has a positive ground truth, and the model predicts a positive outcome for this individual, then, according to the spider diagram, they are both associated with the same individual situated in the intersection of sets $\mathbb{G}$ and $\mathbb{A}$. By extending this analysis to all other individuals in the expected outcome set, the resulting visualization can reveal any potential bias in the AI model. The fairness of the algorithm is assessed using metrics displayed on the bottom right side. In this scenario, the model exhibits fairness towards men but demonstrates bias against women (i.e., in the case of women, the

count of 'x' in the set $\mathbb{G}$ given in the top left is 3, whereas the count of '+' obtained in the intersection region of sets $\mathbb{G}$ and $\mathbb{A}$ is only 2). Further, the degree of bias can be calculated as shown in the diagram (obtained by finding the ratio between the difference of count of 'x' in the set $\mathbb{G}$ and the count of '+' in set $\mathbb{A}$ to the total number of individuals in the subpopulation). The legend box with symbols is in the bottom left corner.

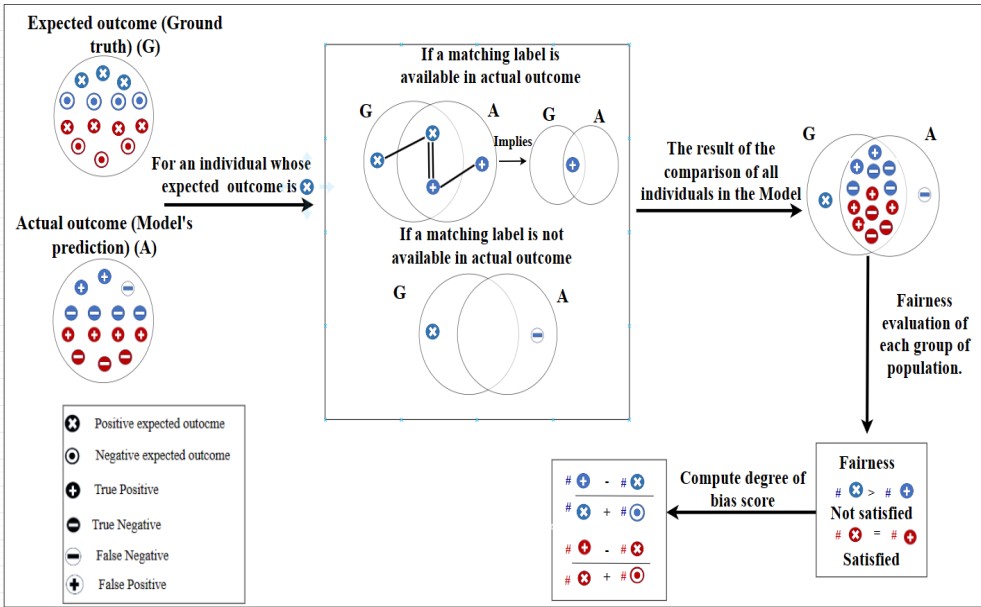

Figure 7: Fairness evaluation using the spider diagram demonstrated using a toy example

## C    CORRECTNESS OF ALGORITHM 1

To check the correctness of the algorithm 1, we use the concept of loop invariants. Here $N$ represents the total number of individuals in the given dataset. $P_0, P_1$, and $P_2$ are the predicates formed from the quantization formula developed for calculating the frequencies of $\alpha$ and $\omega$ classes in both actual and expected outcomes. For example, the predicates are as shown below for the actual outcome.

$$A_\omega = (\#i{:}0{\le}i{\le}N{:}\mathbb{A}[i] = \omega), A_\alpha = (\#i{:}0{\le}i{\le}N{:}\mathbb{A}[i] = \alpha) \tag{2}$$

The above formula calculates the count of the $\omega$ class and $\alpha$ class and stores them in variables $A_\omega$ and $A_\alpha$ respectively. The loop invariant of (2) is $P_0$ and $P_1$ where the constant $N$ is replaced with variable $n$ and then the above equation will become as shown here.

$$P_0{:} A_\omega = (\#i{:}0{\le}i{\le}n{:}\mathbb{A}[i] = \omega), P_1{:} A_\alpha = (\#i{:}0{\le}i{\le}n{:}\mathbb{A}[i] = \alpha) \tag{3}$$

From here we assume the postcondition as $(P_0 \wedge n = N)$. As an initial condition of loop invariant, replace $n$ with 0 in (3) which gives $A_\omega, A_\alpha$=0,0 respectively. This specifies the count of occurrence over the empty range which will always be 0. Hence invariants $P_0$ and $P_1$ hold true for the initial condition. Now we then consider a random case for $n$, where $n \neq N$ and $n > 0$, i.e., a general case where $n = n + 1$. Hence (3) can be rewritten as follows:

$$P_0{:} A_\omega = (\#i{:}0{\le}i{\le}n + 1{:}\mathbb{A}[i] = \omega), P_1{:} A_\alpha = (\#i{:}0{\le}i{\le}n + 1{:}\mathbb{A}[i] = \alpha) \tag{4}$$

To solve this we need to split $i$ from 0 to $n$ and $i$ equals $n$.

$$P_0{:} A_\omega = (\#i{:}0{\le}i{\le}n{:}\mathbb{A}[i] = \omega) + (\mathbb{A}[n] = \omega)$$
$$P_1{:} A_\alpha = (\#i{:}0{\le}i{\le}n{:}\mathbb{A}[i] = \alpha) + (\mathbb{A}[n] = \alpha) \tag{5}$$

This specifies that when the actual output of the model is 1 when the sensitive attribute is in $\omega$ class, our proposed algorithm is incrementing the count of the corresponding variable $A_\omega$, and similarly

if the output of the model is 1 when the sensitive attribute is $\alpha$, the corresponding variable $A_\alpha$ is incremented by 1 which can, in turn, give the count of occurrence of each output class in the actual output of AI model. For doing this consider postcondition $P_2 : \ 0 \leq n \leq N$.

For proving the correctness, consider the case where n,$A_\omega$,$A_\alpha$=0,0,0. We need to check whether all invariants hold true. For that substitute n,$A_\omega$,$A_\alpha$=0,0,0 in $(P_0 \wedge P_1 \wedge P_2)$.

Proof 0 :

$$
\begin{aligned}
&(P_0 \wedge P_1 \wedge P_2) \, (n, A_\omega, A_\alpha := 0, 0, 0) \\
\equiv 0 = &(\#i{:}0\leq i\leq 0{:}f.i = \omega) \wedge \ (\#i{:}0\leq i\leq 0{:}f.i = \alpha) \wedge \ 0 \leq 0 \leq N \\
\equiv 0 = &0 \wedge \ 0 \ \leq N \equiv 0 \leq N
\end{aligned}
\tag{6}
$$

Since $N \geq 0$, we can say that invariance holds true for the initial condition where $n$,$A_\omega$,$A_\alpha$=0,0,0. Now we need to prove for the random case where $n \neq N$. Substitute $n$ as $n+1$ and form the invariant as shown below.
proof 1 :

$$
\begin{aligned}
&P_0 \ \wedge \ P_1 \ \wedge P_2 \wedge n \neq N \\
\equiv \ &(\#i{:}0\leq i\leq n+1{:}f.i = \omega)\wedge (\#i{:}0\leq i\leq n+1{:}f.i = \alpha) \ \wedge \ (0 \leq n+1 \ \leq N) \\
\equiv \ &(A_\alpha + (f.n = \alpha)) \ \wedge \ (A_\omega + (f.n = \omega)) \wedge (n+1 < n) \implies \ n \neq N
\end{aligned}
\tag{7}
$$

This clearly shows that the invariant will hold true for any random case where $n = n + 1$. At last for the termination case where $n = N$, substitute $n$ as $N$ in the predicates $P_0$, $P_1$ and $P_2$ and prove the equation as shown below.
Proof 2 :

$$
\begin{aligned}
&(P_0 \ \wedge \ P_1 \ \wedge \ P_2) \ \wedge (n = N) \\
\equiv \ &(\#i{:}0\leq i\leq N{:}f.i = \omega) \wedge (\#i{:}0\leq i\leq N{:}f.i = \alpha) \wedge (0 \leq N \leq N)
\end{aligned}
\tag{8}
$$

Hence termination condition also holds.
Similarly, for the expected outcome the predicates will be as shown below:

$$
G_\omega = (\#j{:}0\leq j\leq N{:}\mathbb{G}[j] = \omega), G_\alpha = (\#j{:}0\leq j\leq N{:}\mathbb{G}[j] = \alpha)
\tag{9}
$$

The above equation counts the occurrence frequency of $\omega$ class and $\alpha$ class in the expected outcome (output) of an AI model and stores it in variables $G_\omega$ and $G_\alpha$ respectively. For doing this we need to form the invariants $P_0, P_1,$ and $P_2$ as shown below:

$$
\begin{aligned}
&P_0{:}\ G_\omega = (\#j{:}0\leq j\leq N{:}\mathbb{G}[j] = \omega) \\
&P_1{:}\ G_\alpha = (\#j{:}0\leq j\leq N{:}\mathbb{G}[j] = \alpha) \\
&P_2{:}\ 0 \leq n \leq N
\end{aligned}
\tag{10}
$$

Proof of correctness for (10) remains the same as above. To determine the extent of bias in an AI model for both the $\alpha$ and $\omega$ classes, you can compute bias scores using the following formulas:

For the $\alpha$ class, the bias score is calculated as: $d_\alpha = \frac{(A_\alpha - G_\alpha)}{(\alpha)}$

For the $\omega$ class, the bias score is calculated as: $d_\omega = \frac{(A_\omega - G_\omega)}{(\omega)}$

These formulas utilize the values obtained from the previous steps, where $G_\omega$ and $G_\alpha$ represent expected outcomes, and $A_\omega$ and $A_\alpha$ represent actual outcomes. These bias scores help quantify the degree of bias present in the AI model for each respective class, taking into account the number of instances in each class denoted by $(\#\alpha)$ and $(\#\omega)$.

## D  EXPERIMENTAL DETAILS

In this work, we take five datasets – social network ads prediction, loan approval prediction (dat, c), German credit score prediction (dat, b), UCI adult (dat, a) and US faculty hiring (dat, d) (Wapman et al., 2022) to visualize the bias and to verify fairness in the AI model. The initial step involves data preprocessing, where certain features are eliminated from the dataset. Afterwards, discrete and

discretized continuous attributes are encoded using label encoding. The datasets used in this paper is obtained from the Kaggle website. For creating the model, first we identified $\alpha$ and $\omega$ classes based on the sensitive attributes in the dataset. We then recorded the expected outcome of the model using Euclidean distance.

Table 3: Datasets used in this paper

| Dataset | Dataset Number | Target Feature | #(Instances) | Test % |
|---|---|---|---|---|
| Social Network | D1 | Purchased | 400 | 20 |
| Loan approval | D2 | Loan-status | 981 | 20 |
| German credit | D3 | Risk | 1000 | 20 |
| UCI adult | D4 | Salary | 104099 | 30 |
| US faculty hiring | D5 | #(Men, Women) | 411544 | 30 |

## E  PERFORMANCE METRICS DETAILS

The following is the list of performance metrics for binary classification mentioned in this paper in section 3.4.

**Accuracy**$= \frac{TP+TN}{TP+FP+TN+FN}$

**Sensitivity**$= \frac{TP}{TP+FN}$

**Specificity** $= \frac{TN}{TN+FP}$

**Precision** $= \frac{TP}{TP+FP}$

We refer interested readers to refer (Raschka, 2014) for further details.

## F  ADDITIONAL EXPERIMENTS

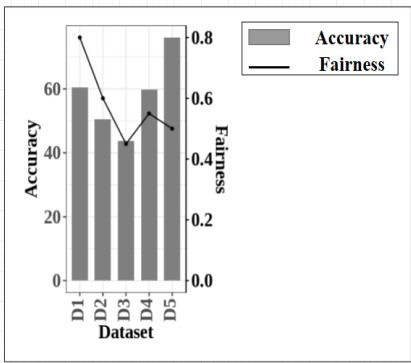

Figure 8: Demonstrates observable tradeoff between accuracy and fairness.

Figure 8 illustrates the relationship between accuracy and fairness calculated across 5 distinct datasets. Each bar in the figure corresponds to the accuracy of a specific dataset, while the connecting lines depict the fairness values. The result suggests a correlation where fairness tends to align with accuracy.

In this section, we present the additional experiment results. In section 4, we use a stacked bar chart to present the results of the spider diagram for enhanced visualization. In this particular section, we offer a detailed depiction of bias using spider diagrams obtained from the program. The sets in each sub-figure represent the group of $\alpha, \omega$, $\mathbb{G}$ (Expected outcome), and $\mathbb{A}$ (Actual outcome) for various models. These sets are divided into zones represented using different colors and the number inside each zone (intersection region of corresponding sets) indicates the count of spiders (instances) residing in it. As an example, in Figure 9a there are 96275 *false positives* in the model out of which 20753 are *Female* ($\alpha$) and 75522 are *Male* ($\omega$). The intersection region of sets $\mathbb{G}$ and $\mathbb{A}$ represents

the count of instances having Correct Prediction (CP that includes both TP and TN). The region inside the zone $\mathbb{G} \setminus \mathbb{A}$ represents the count of False positives (FP) in the model and the zone $\mathbb{A} \setminus \mathbb{G}$ represents False Negative (FN) of the model.

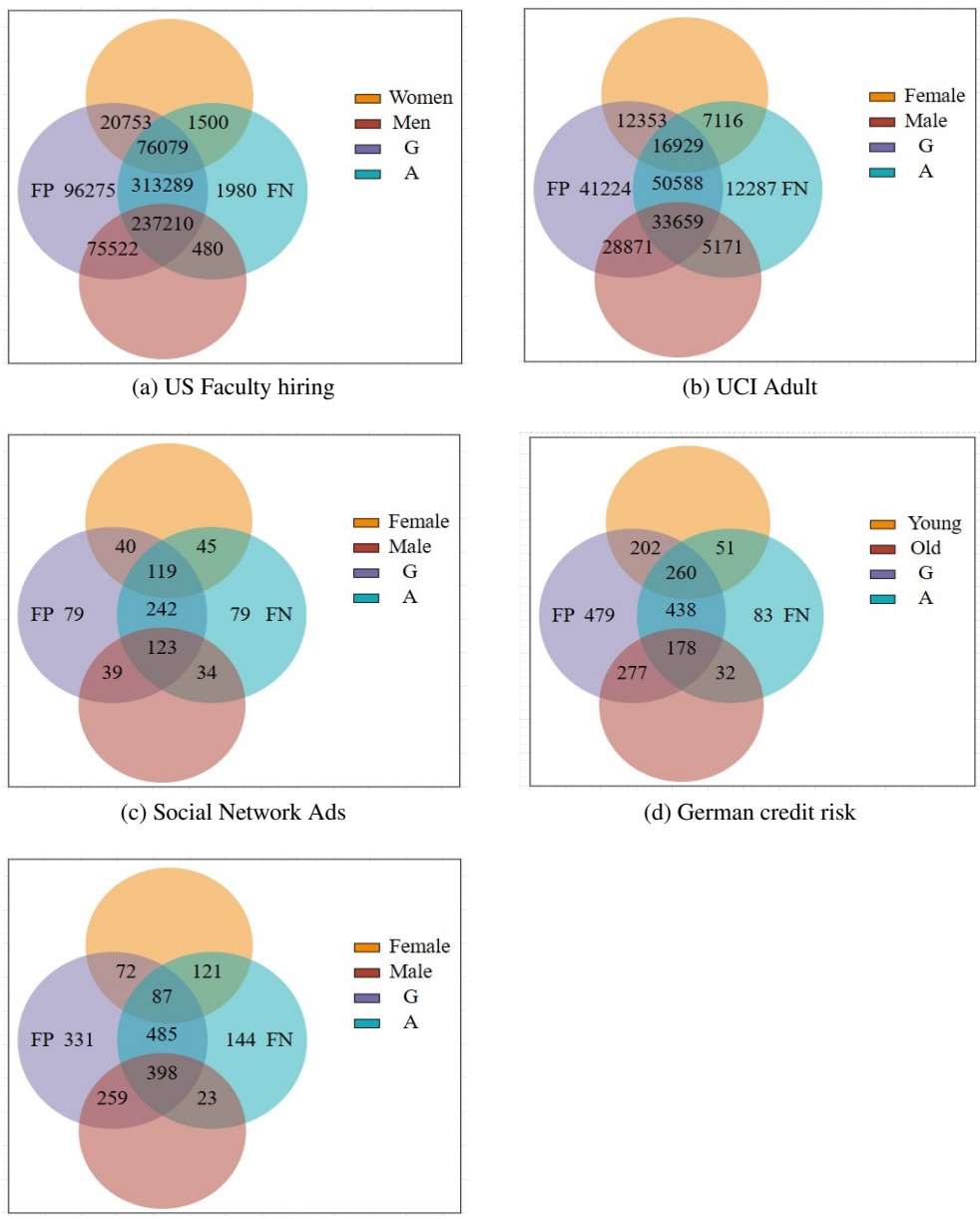

Figure 9: Bias visualization of various datasets using spider diagram. Number inside each zone indicates the number of spiders (dataset instances) residing in it.

