# OpenReview forum: "A Logical Framework for Verification of AI Fairness"
_ICLR.cc/2024/Conference — Submitted to ICLR 2024_

### Official Review · Reviewer_Wzut · 2023-10-31

**Soundness:** 3 good
**Presentation:** 2 fair
**Contribution:** 1 poor
**Rating:** 5
**Confidence:** 4

**Summary:**

This paper is devoted to analysing ML model fairness and proposes a simple
approach to determine whether (and how much) a model is biased towards/against
some groups of entities based on so-called spider diagrams. The idea is as
follows: given a dataset, one can compute the ratio of discrepancy between the
expected outcomes and actual outcomes reported by the model. If one
additionally measures similarity between various entities in the dataset, such
discrepancy observed for similar entities gives us an indication that a bias
is present. The authors claim a few theoretical results and propose an
algorithm that computes the "degree of bias" and evaluate their ideas
experimentally. Experimental results demonstrate that the proposed approach is
computationally more efficient than confusion matrices.

**Strengths:**

- The paper seems clearly written. The ideas are simple and, as a result, easy
  to follow.
- Visualization of fairness / bias based on spider diagram looks nice.
- The proposed approach works faster than the alternative based on confusion
  matrices.

**Weaknesses:**

- The first weakness is intertwined with one of the strengths, which is the
  simplicity of the ideas. Unless I overlook something important, they look
  too plain for a conference of this level. I mean computing the discrepancy
  ratio between the actual outputs of the model and what is expected based on
  the dataset and our (heuristic) similarity measure seems rather
  straightforward to me.

- The paper does not argue why this measure of fairness is valuable. For
  instance, it is unclear to me what happens in the case when a dataset we
  start from is biased on its own. There should be a way to alleviate this by
  sacrificing model accuracy but the authors do not discuss this nor they say
  how data bias affects their fairness measure.

- The paper fails to relate with the state of the art in fairness analysis,
  inclding previous works on the use of logic. For example, these papers and
  references therein:

  [A] Alexey Ignatiev, Martin C. Cooper, Mohamed Siala, Emmanuel Hebrard, João
  Marques-Silva: Towards Formal Fairness in Machine Learning. CP 2020: 846-867

  [B] Ulrich Aïvodji, Julien Ferry, Sébastien Gambs, Marie-José Huguet,
  Mohamed Siala: FairCORELS, an Open-Source Library for Learning Fair Rule
  Lists. CIKM 2021: 4665-4669

  [C] Julien Ferry, Ulrich Aïvodji, Sébastien Gambs, Marie-José Huguet,
  Mohamed Siala: Improving fairness generalization through a sample-robust
  optimization method. Mach. Learn. 112(6): 2131-2192 (2023)

- In my view, the presented experimental results are rather weak - the speedup
  on N milliseconds compared to M milliseconds does not look important. The
  authors argue that they reduce the number of function calls but I fail to
  see why this is significant to them if the performance of the tool is only
  slightly better than that of the competitor. There is no discussion /
  comparison of the proposed metric and the corresponding approach in terms of
  the quality of the produced fairness assessment.

**Questions:**

- How is your fairness metric affected by dataset (not model) bias?

- How does your work relate with state of the art in logic-based fairness
  analysis?

- What function calls are meant here? Why are they important?

---

> ### Author Response · Authors · 2023-11-22
>
> 1.How is your fairness metric affected by dataset (not model) bias?
>
> - We have showcased the correlation between accuracy and fairness resulting from our methodology in Appendix F on page 17 (Figure 8).  We have added an explanation as follows: “Figure 8 illustrates the relationship between accuracy and fairness calculated across 5 distinct datasets. Each bar in the figure corresponds to the accuracy of a specific dataset, while the connecting lines depict the fairness values. The result suggests a correlation where fairness tends to align with accuracy.” (Appendix F, Para 1)Our findings suggest a close relationship between accuracy and fairness when employing our approach to detect bias within an AI model. While dataset bias often necessitates a tradeoff between accuracy and fairness, our current study predominantly addresses model bias rather than dataset bias. However, we acknowledge the importance of investigating dataset bias as a potential future avenue, as highlighted in the second paragraph of our conclusion “ Adding to this, the study can be extended to investigate accuracy- fairness tradeoff in case of dataset bias”.
>
> 2. How does your work relate with state of the art in logic-based fairness analysis?
>
> - It may be noted that  current logic-based fairness analysis evaluates the fairness of models through sensitive attributes
> (Counterfactual fairness, FTU). In our work, we emphasize statistical measures for this purpose. Please find the below-mentioned
> references:
>   1. Alexey Ignatiev, Martin C. Cooper, Mohamed Siala, Emmanuel Hebrard, João Marques-Silva: Towards Formal Fairness in
>    Machine Learning. CP 2020: 846-867
>   2. Kusner, Matt J., Joshua Loftus, Chris Russell, and Ricardo Silva. "Counterfactual fairness." Advances in neural information
>     processing systems 30 (2017).
>
>
> 3. What function calls are meant here? Why are they important?
>
> - We have added clarification of this matter in (section 4 page 8 para 2)."Here the number of recursion or function calls can be crucial in assessing a model's performance for a few reasons. Firstly, it indicates the computational load and efficiency of the model. A high number of recursive calls can suggest increased computational complexity, potentially leading to longer processing times or resource- intensive operations." Additionally, we are providing references that emphasize the need for function calls and explain the importance of employing
> optimization methods.  These references are mentioned in the draft also.
>
>    1. Ousterhout JK. Optimizing Program Performance
>
>    2. Grzeszczyk, M.K., 2018. Optimization of Machine Learning Process Using Parallel Computing. Advances in Science and
>        Technology. Research Journal, 12(4)
>
>    3. Nima Asadi, Jimmy Lin, and Arjen P De Vries. Runtime optimizations for tree-based machine
>        learning models. IEEE Transactions on Knowledge and Data Engineering, 26(9):2281–2292, 2013.

---

### Official Review · Reviewer_cYyx · 2023-11-01

**Soundness:** 3 good
**Presentation:** 1 poor
**Contribution:** 2 fair
**Rating:** 3
**Confidence:** 2

**Summary:**

This paper develops an approach to evaluating AI fairness using spider diagrams, a visualization of monadic first-order logic with equality based on Venn diagrams. Experiments are done showing that the current method is superior to some existing (confusion matrix) in terms of processing time and function calls required.

**Strengths:**

The approach and the use of spider diagrams is novel. The problem is timely and well-situated within the literature.

**Weaknesses:**

I cannot clearly understand the contribution from the paper as currently written. I'm not an expert in the area of fairness and no doubt this is part of the reason, but I also think the presentation has a lot of issues.

Section 2.1 introducing the diagrams is not clear. Please expand, including formal definitions and (especially) informal examples. Just adding the note that this is equivalent to monadic FOL with equality would be really helpful. I expect that this is not going to be familiar to most (including myself); I had to consult external references, and this should really be self-contained.

Can the authors simply use first-order logic instead? This is going to be familiar to a lot more readers. I do not understand what about the approach relies on spider diagrams specifically. E.g., is it claimed that they are more intuitive? Then there should be an example showing how they add to that. I saw that Appendix E just uses Venn diagrams, there is no need to add spiders or anything else.

**Questions:**

- What is phi in Theorem 1? Is this the psi from semantics for spider diagrams? Needs to be self-contained

Minor comments:
- page 3: "where each instance is a tuple ..." In the tuple, "yhat in 0, 1" should be "yhat in {0, 1}"
- Definition 2: forall quantifier in S should probably just be in the set, i.e., {(e_i, a_i): e_i in E, a_i in A, i = 1, ..., N}

---

> ### Author Response · Authors · 2023-11-22
>
> 1. What is phi in Theorem 1? Is this the psi from semantics for spider diagrams? Needs to be self-contained
>
>      - Phi is from the semantics of the spider diagram which basically means a zone outside the contours (sets). Please see the reference for further details.
>
>      - Gem Stapleton, John Taylor, Simon Thompson, and John Howse. The expressiveness of spider diagrams augmented with
>        constants. Journal of Visual Languages & Computing, 20(1):30–49, 2009.
>
> 2. Please expand, including formal definitions and (especially) informal examples. Just adding the note that this is equivalent to monadic FOL with equality would be really helpful.
>
>       - We have added "Spider diagrams are higher-order representations of Venn diagrams and are equivalent to monadic FOL with
>        equality" in the draft (subsection 2.1, para 1).
>
> 3. I do not understand what about the approach relies on spider diagrams specifically. E.g., is it claimed that they are more intuitive? Then there should be an example showing how they add to that.
>
>       - We have included a demonstration that shows the working of our method using a toy example in Appendix B  page 15 (Figure 7).
>
> 4. page 3: "where each instance is a tuple ..." In the tuple, "yhat in 0, 1" should be "yhat in {0, 1}"
>
>      - Thank you for bringing this to our attention. We have changed in the draft.
>
> 5. Definition 2: forall quantifier in S should probably just be in the set, i.e., {(e_i, a_i): e_i in E, a_i in A, i = 1, ..., N}
>
>      - Thank you for bringing this to our attention. We have changed in the draft.

---

> > ### Comment · Reviewer_cYyx · 2023-11-23
> >
> > Thanks for the response. I maintain my current rating. I am still not able to grasp the approach and contribution completely.

---

### Official Review · Reviewer_iDyr · 2023-11-05

**Soundness:** 1 poor
**Presentation:** 1 poor
**Contribution:** 1 poor
**Rating:** 1
**Confidence:** 4

**Summary:**

The paper proposes a logical framework  "FairAI" based on "spider"  - a generalisation of the Venn diagrams- as an alternative fairness metrics (alternative to equalised odds, statistical disparity etc,) , and experimentally show that their approach is   by large more performant compared to previous approaches in terms of function calls and performance times.

**Strengths:**

My apologies but I am unable to list any.

**Weaknesses:**

- Not clear which AI model that the authors use. (some AI model based on an ArXiv paper)

- How is the threshold chosen (well average), and why such expected outcome should behave nicely across all groups is not clear..

-Counter-factual/Causal fairness  metrics  are totally disregarded.

- Exposition has so many flaws, even if the results were significant, in its current form it would be hard to justify that it should be published.

-  Theorems are almost trivial, and I don't see any "verification" to be honest. Overall  I have strong doubts about the correctness of the approach, let alone significance of the results.

**Questions:**

I don't have any.

**Details Of Ethics Concerns:**

I don't have any.

---

> ### Author Response · Authors · 2023-11-22
>
> 1. Counter-factual/Causal fairness metrics are totally disregarded.
>
> - This paper primarily focuses on statistical fairness. We have added clarification of this matter in Section 1 para 2 as follows:
> "In the realm of evaluating fairness in an AI model, there are multiple approaches. These include statistical measures, individual fairness considerations, Fairness Through Unawareness (FTU), counterfactual or causal fairness, and logic-based approaches. It's important to note that in the case of counterfactual fairness, a scenario where, for instance, the gender of an individual is hypothetically changed to a different value would lead to differences in other features as well. This complexity arises due to the interconnected nature between sensitive and non-sensitive attributes, making it challenging to accurately assess bias. Likewise, in the case of Fairness Through Unawareness (FTU), when certain features are linked or correlated with sensitive attributes, a model that overlooks these sensitive features doesn't guarantee fairness (Castelnovo et al., 2022). Hence this paper primarily focuses on statistical fairness criteria. "
>  In this approach, individual fairness will be spoiled as these methods don't depend on the actual scenario (ground truth) to evaluate fairness.
>
> 2. Theorems are almost trivial, and I don't see any "verification" to be honest
>
> - Please note that understanding the theorems becomes more straightforward when there's a clear understanding of spider diagrams, their application in visualizing bias, and confirming fairness. We perceive it as a strength rather than a weakness.

---

### Official Review · Reviewer_VPjM · 2023-11-09

**Soundness:** 1 poor
**Presentation:** 1 poor
**Contribution:** 2 fair
**Rating:** 1
**Confidence:** 2

**Summary:**

This paper proposes a novel framework for visualizing and verifying the fairness of AI using spider diagrams.

Unfortunately, I was not able to understand some very crucial and important aspects of the paper despite several attempts. This is not helped by several typos and grammatical errors, including using the singular form where perhaps the authors meant to use the plural form, making it very confusing to read. I found myself constantly trying to guess what the authors mean to say. I am happy to revise my review if the authors can help clarify things. I will try my best to state how I interpreted the paper, in the hope that the authors can jump in and help clarify if I get something wrong.

The authors consider a setting where we are given a dataset D. Each instance in the dataset has an actual label belonging to the set A. An AI model M (a function) maps each instance to a label, where the expected label for an instance belongs to the set E. The instances in D, together with their actual and expected labels are used to create a spider diagram. Each spider represents an instance. My understanding is that for each instance i, there is a spider, and the feet of the spider represent its actual or expected label, with a tie connecting the feet for the same instance. Now, if an instance i has a foot in the intersection of A and E, it means M correctly labels instance i. The degree of bias of M is described by comparing across the different classes (e.g. of a protected attribute) the frequencies of spiders (corresponding to instances of each class) that do not have a foot in the intersection of the sets E and A in the spider diagram.

Overall, I think the paper would benefit greatly from the addition of simple toy examples to illustrate the usefulness of the proposed framework. I suggest adding an example of a binary classification task, with a dataset D with a small number of instances, with the actual labels, a simple biased model, and expected labels described clearly, and showing step by step how the spider diagram helps illustrate the bias of the model.

**Strengths:**

- The use of spider diagrams and the proposed logical framework appears to be novel if it is sound. Unfortunately, I was unable to verify this.
- If sound, the proposed approach appears to be a promising visualization tool to identify bias.

**Weaknesses:**

The following are weaknesses in either the technical aspects or presentation of the paper, which if addressed may make the paper easier to understand. I will try to list them as they are encountered while reading the paper from the beginning.
- In the abstract, it is not clear what is meant by actual outcome and expected outcome. Specifically, the phrases used are "actual outcome of the algorithm" and "expected outcome of the model". Is the actual outcome determined by an algorithm, or does it refer to some ground truth about the instance (say, determined by a target function)? It is actually not clear what is meant by model here. Is it a set of hypotheses, or is it a single hypothesis function? What is meant by expected outcome of the model? Does the model describe a probability distribution over possible labels? Or is it that depending on the available dataset (generated from some input distribution), a different function is learned? What is meant by algorithm and model here? How are they different? By model, do you mean the nearest-neighbor / similarity based method described in Section 2, Page 3?
- I suggest changing the notation of the set of expected outcomes, as \mathbb{E} is typically used to represent the expected value of a random variable.
- Page 2, para 2: "compares the set of expected outcome E to the set of actual outcome A". Do you mean to use the singular or plural here? e.g. set of expected outcomes. In a binary classification task, what are E and A?
- Page 2, Section 2: "two groups of output"? Do you mean protected attribute values? Typically the output refers to the predicted label. What am I missing?
- Page 3, para 1: "protected groups are advantaged ..." Is this an assumption, or a requirement? It is not clear what is meant here.
- Page 3, para 2: "generator": Is denoted by p, but then does not appear in Eq. (1). Is Q_1 the same as p? The sensitive attribute is denoted s, but then is not mentioned or discussed later in the paper.
- There is also a claim here: "If two entities Q_1 and Q_i are similar, the expected outcome of both ... should be classified into the same class label depending on the class of Q_1" This reads like a very strong assumption about the problem setting. Consider e.g. the setting where there is a single integer attribute, and all instances with odd value for the attribute have ground truth label 0 and all even instances have ground truth label 1. How do we handle such a problem?
- In Eq. (1), what does 'n' refer to? Earlier, 'n' was used as a variable to index the instances. Here, its use seems different.
- What is the threshold for deciding whether the similarity between Q_1 and Q_i is sufficient to assign Q_i the same label as Q_1?
- Page 3, last 2 lines: "each closed curve is used to represent sets": Do you mean multiple sets or a single set? If multiple, how to intepret this statement?
- I did not find the discussion of Section 2.1 to be helpful. Referring back to the original papers by Howse et al. helped clarify some things, and I can see how spider diagrams are useful for logical reasoning, but I am unable to completely understand its use in evaluating an AI model. An example that builds from a toy AI problem with a small dataset and a simple biased model would be greatly appreciated.
- Definition 1. Do you mean to say for each expected label e_i, there exists an actual label a_i, such that e_i = a_i? Could you illustrate how this works using the example of a binary classification problem? Can an instance i have multiple expected and actual labels? Is it possible for an instance to have an expected label but no actual label or vice-versa?
- With a few assumptions, I can possibly see how in Section 3, the proposed algorithm can be used to compute the degree of bias. However, it would help to clarify the presentation and provide a running example to remove any ambiguities.
- Figure 3 could be used to show the actual spider diagrams in addition to the bar plots.

**Questions:**

Please see the questions in the comments above.

---

> ### Author Response · Authors · 2023-11-22
>
> 1.In the abstract, it is not clear what is meant by actual outcome and expected outcome. Specifically, ...
>
> - We have changed the phrase to "This framework compares the sets representing the actual outcome of the model and the expected outcome to identify bias in the model" (Abstract). "The expected outcome (ground truth) of the model is obtained by considering the similarity score between the individuals (how much alike the elements are, calculated based on the distance between the values of corresponding features). The actual outcome is the outcome given by the model's prediction" (Section 1 para 3)”. Model in this paper refers to a classification model and the definition is given in section 2 para 1.
>
> 2. I suggest changing the notation of the set of expected outcomes, as \mathbb{E}.
>
> - We have changed the notation to "\mathbb{G}".
>
> 3. Page 2, para 2: "compares the set of expected outcome E to the set of actual outcome A".
>
> - We have changed to  " To verify fairness in the model, the framework compares the set of the expected outcomes ("\mathbb{G}) to the set of the actual outcomes ("\mathbb{A})" (section 1 para 3).
>  "In this paper, we introduce the notations \mathbb {G} and \mathbb{A} to denote the sets of expected and actual outcomes respectively, produced by an AI model". (section 2 para 1)
>
> 4. Page 2, Section 2: "two groups of output"?
>
> - The term “two groups of output” was initially used to represent two demographic groups in the model. In the revised draft, we have changed to "In this study, we use an AI model defined by Das & Rad (2020) with two demographic groups (subpopulation)—non-protected (α) and protected (ω)—based on the sensitive attribute(s)" for better clarity (Section 2 para1).
>
> 5. Page 3, para 1: "protected groups are advantaged ..." Is this an assumption...
>
> - In an AI model, based on case studies such as COMPAS and Amazon’s recruitment algorithms, there exist two or more demographic subpopulations that can influence a model’s outcomes. The group for which the model's outcome aligns favorably is termed the "protected group," while the group where the outcome is unfavorable or doesn't align with the ground truth is termed the "non-protected group." In this scenario, we can describe the protected groups as benefiting from the model's predictions. This is an assumption that aligns with past experiences. (Section 2 para 1) (Case studies are referenced in the introduction of this work (Section 1 para 1)).
>
> 6. Page 3, para 2: "generator": Is denoted by p...
>
> - The symbol 's' denotes the sensitive attribute.  To avoid confusion, the sentence is rephrased as follows:" Let $a_1, a_2,\ldots,a_m$ be the attributes, that include both sensitive attributes (i.e. race, ethnicity, sex) and non-sensitive attributes in the model" (section 2 para 2). Q_1 is the same as p. We have changed in the draft.
>
> 7. There is also a claim here: "If two entities Q_1 and Q_i ...
>
> -  If a specific condition, like the one described earlier, helps the classification process for individuals, the proposed approach can be applied to ground truth value and categorize individuals. In this context, for a single integer attribute, we can compute the Euclidean distance between two points. Depending on this distance value, if it's odd, the individual is classified into class 0; otherwise, they're classified into class 1.
>
> 8. In Eq. (1), what does 'n' refer to?
>
> - We have changed 'n' to 'm'. Here 'm' denotes the total number of attributes in the model.  (Section 2 para 2 page 3).
>
> 9. What is the threshold for deciding whether the similarity...
>
> - The average value of Euclidean distance between the two individuals. (Discussed in page 3 para 3)
>
> 10. Page 3, last 2 lines: "Each closed curve is used to represent sets"...
>
> - We have changed to " Spider diagrams are higher-order representations of Venn diagrams and are equivalent to monadic FOL with equality. Here each closed curve is used to represent a set and is labeled and enclosed in a rectangle." in the draft (Section 2.1 para 1).
>
> 11. I did not find the discussion of Section 2.1 to be helpful...
>
> - We have included a demonstration that shows the working of our method using a toy example in Appendix B  page 15 (Figure 7).
>
>
> 12. Definition 1. Do you mean to say for each expected label...
>
> - An instance "i" can have only one expected outcome (i.e., ground truth) and one actual outcome (i.e., predicted outcome). This is demonstrated with a toy example using a small example dataset (Appendix B  page 15 (Figure 7)).
>
> 13. With a few assumptions, I can possibly see how in...
>
> - We have included a demonstration that shows the working of our method using a toy example in Appendix B  page 15 (Figure 7). It demonstrates the method used in calculating the degree of bias.
>
>
> 14. Figure 3 could be...
> - Appendix E, figure 9 presents the actual spider diagrams illustrating bias visualization across five datasets, in addition to the information provided in Figure 3.

---

### Author Response · Authors · 2023-11-22

We would like to thank the reviewers for their detailed comments and suggestions and the area chairs for giving us an opportunity to revise our manuscript. We have revised the manuscript and we believe this thoroughly revised version addresses all the comments made by the reviewers. Here, we indicate the changes we made to our manuscript to address the review comments and suggestions.

---

### Meta-Review · Area_Chair_cjJ8 · 2023-11-30

**Metareview:**

This paper proposes a logic framework based on spider diagrams for defining and verifying fairness in AI.

Overall, the reviewers agree that the paper is addressing an important problem in fair AI, and the use of spider diagram seems novel.
However, the reviewers also agree that the paper is not yet ready for publication, highlighting several areas for enhancement. The primary concerns revolve around its presentation, with many reviewers finding it challenging to grasp the work and its contributions fully. Enhancing the paper's clarity, perhaps through the inclusion of straightforward, informal examples, and more explicitly outlining its contributions in comparison to standard fairness literature could substantially improve it. We hope the authors find the review comments helpful in improving their manuscript.

**Justification For Why Not Higher Score:**

The current presentation are not up to the bar for publication yet.

**Justification For Why Not Lower Score:**

N/A

---

### Decision · Program_Chairs · 2024-01-16

Reject